# Super-Low Friction Electrification Achieved on Polytetrafluoroethylene Films-Based Triboelectric Nanogenerators Lubricated by Graphene-Doped Silicone Oil

**DOI:** 10.3390/mi14091776

**Published:** 2023-09-16

**Authors:** Junzhao Chen, Yu Zhao, Ruirui Wang, Pengfei Wang

**Affiliations:** Institute of Nanosurface Science and Engineering (INSE), State Key Laboratory of Radio Frequency Heterogeneous Integration, College of Mechatronics and Control Engineering, Shenzhen University, Shenzhen 518060, China; chenjunzhao1998@163.com (J.C.); zy662042@163.com (Y.Z.); wrr571523957@163.com (R.W.)

**Keywords:** triboelectric nanogenerator, super-low friction, triboelectrical performance, PTFE film, oil lubrication, graphene nanosheets

## Abstract

The novel proposal of Wang’s triboelectric nanogenerator (TENG) has inspired extensive efforts to explore energy harvesting devices from the living environment for the upcoming low-carbon society. The inevitable friction and wear problems of the tribolayer materials become one of the biggest obstacles for attaining high-performance TENGs. To achieve super-low friction electrification of the TENGs, the tribological and electrical behaviors of the sliding-mode TENGs based on polytetrafluoroethylene (PTFE) films and metallic balls under both dry friction and liquid lubrication conditions were investigated by using a customized testing platform with a ball-on-flat configuration. Most interestingly, a super-low friction coefficient of 0.008 was achieved under graphene-doped silicone oil lubrication. The corresponding wear rate of the PTFE film was drastically decreased to 8.19 × 10^−5^ mm^3^/Nm. Simultaneously, the output short-circuit current and open-circuit voltage were enhanced by 6.8 times and 3.0 times, respectively, compared to the dry friction condition. The outstanding triboelectrical performances of the PTFE film when sliding against a steel ball are attributed to the synergistic lubricating effects of the silicone oil and the graphene nanosheets. The current research provides valuable insights into achieving the macro-scale superlubricity of the TENGs in practical industrial applications.

## 1. Introduction

Since the 21st century, clean energies such as solar, bioenergy, wind, and ocean power have rapidly developed, opening the era of green energy for human beings. In 2012, Wang’s research group first invented an energy harvesting device called a triboelectric nanogenerator [1]. It is based on the principles of electrostatic induction and contact electrification, capable of efficiently converting mechanical energy from the living environment (sources include human motion, ocean waves, and wind) into electricity or signal. Compared with traditional energy harvesting devices, TENGs have many advantages, including small size, light weight, high voltage output, versatile shapes, and high compatibility [2,3,4], which making them good candidates for advanced cutting-edge applications in self-powered sensors, blue energy harvesting, and micro-nano energy scavenging fields [5,6,7,8].

During the operation of TENGs, friction and wear occur due to the periodic contact or sliding between two materials with opposite electron affinities [9]. In case of the solid-solid contact scenarios, the frictional surfaces of the materials might experience severe wear, affecting not only the stability of electrical outputs but also the TENG’s lifespan and potential applications [10]. To address the issue of frictional wear in TENGs, researchers have proposed various approaches, such as modifying polymer materials [11,12,13], transforming mechanical structures from sliding to rolling friction [14,15,16], and incorporating lubricants [17,18,19,20]. These great efforts have yielded promising results, such as reducing friction coefficient, enhancing wear resistance, extending the TENG’s lifespan, and improving electrical output.

The introduction of liquid lubricants to the sliding interfaces has proven to be a simple and effective approach for minimizing frictional wear in the industrial fields [21,22,23]. Moreover, the addition of two-dimensional (2D) materials (e.g., graphene [24,25], molybdenum disulfide [26], and layered double hydroxide nanoplatelets [27]) to lubrication oils could enhance the load-bearing capacity and wear resistance to further mitigate the frictional wear [28,29]. Therefore, it can be assumed that the addition of lubrication oils in the TENGs not only could promote tribological behavior and mechanical durability but also could enhance the electrical output performance. Wu et al. [30] found that appropriate liquid lubrication of TENGs could significantly improve the wear resistance and the electrical outputs of the TENG. Wear could hardly be detected on the worn surface of the polymer tribolayer even after 36,000 sliding cycles. Wang et al. [31] clarified that adding hexadecane on the nylon surface could cause a 73% reduction in the friction coefficient and an over 50 times increase in the maximum conversion efficiency. Chung et al. [32] designed a nonpolar liquid submerged TENG with a rolling electrode and it was found that even after 72 h of continuous operation no visible damage could be observed on the aluminum tribolayer. Liu et al. [33] studied the effect of various liquid lubricants on the triboelectrical performance of TENG with a ball-on-flat configuration. It was declared that appropriate lubricants as well as liquid volume could enhance the electrical output and achieve long-term stability of the TENG. Moreover, the addition of onion-like carbon (OLC) to nonpolar lubrication oil could create a “micro-bearing” effect where some OLC particles roll between the interfaces to further improve the triboelectrical performance of the TENG [34].

Interestingly, Huang et al. [35] successfully implemented a structural Schottky superlubric triboelectric nanogenerator (SL-TENG) at the micro-scale, making it possible to generate triboelectricity under superlubricity (i.e., friction coefficient of lower than 0.01). Zhang et al. [36] fabricated a macro-superlubric triboelectric nanogenerator with a super-low friction coefficient (less than 0.01) and an ultra-low wear rate based on the excellent hydrogenated diamond-like carbon (H-DLC) films. These pioneering works shed new light on exploring the TENGs with superior tribological behaviors (super-low friction coefficient and wear rate) and outstanding electrical performances (high output short-circuit current and open-circuit voltage).

In this study, to achieve super-low friction electrification of the TENGs, the tribological and electrical behaviors of the sliding-mode TENGs based on the polytetrafluoroethylene (PTFE) films and metallic balls under both dry friction and liquid lubrication conditions were systematically explored by using a customized testing platform with a ball-on-flat configuration. Particularly, the super-low friction coefficient of 0.008 and low wear rate of 8.19 × 10^−5^ mm^3^/Nm were achieved with the PTFE film running against a steel ball under a graphene-doped silicone oil lubrication condition. Simultaneously, the output short-circuit current and open-circuit voltage were enhanced by 6.8 times and 3.0 times, respectively, compared to the dry friction condition. The underlying mechanisms for the excellent triboelectrical performances of the PTFE films-based TENGs were clarified.

## 2. Materials and Methods

PTFE films and metallic balls were selected as the dielectric tribolayer materials. Commercially available PTFE film with a thickness of 0.2 mm was procured from the Daoguan Rubber Products Company (Shanghai, China) and cut into dimensions of 30 mm × 30 mm for the sliding tests. The surface roughness of the pristine PTFE film was measured by using a surface profiler (SJ-210, Mitutoyo, Kawasaki, Japan), as shown in Appendix A. The average surface roughness (R_a_) of the pristine PTFE film from five different locations, as listed in Appendix A, was calculated to be 0.193 μm. Three types of commercial metallic balls (stainless-steel ball (SUS304), copper ball (H62), and aluminum ball (1060)) with diameters of 6.35 mm were purchased from the Yueli Hardware Products Company (Yiwu, China) and used in this study. The metallic balls and PTFE films were ultrasonically cleaned with ethanol and acetone successively for 20 min each by using an ultrasonic bath (M2800-C, Branson, Shanghai, China).

Figure 1a illustrates a self-developed testing platform which could simultaneously measure the friction coefficient, short-circuit current, and open-circuit voltage of the PTFE films running against the metallic balls under the dry friction and liquid lubrication conditions. The photograph of the testing platform is shown in Appendix A. It mainly consisted of a horizontal ball-on-flat reciprocating sliding tribometer, where the PTFE film was driven to slide against the stationary metallic ball by using a linear motor. Detailed information about the system could also be found in Refs. [37,38]. The schematic diagram of the upper and lower tribopair of the sliding-mode TENG is shown in Figure 1b. Specifically, the metallic ball as the upper tribopair was fixed on the upper holder by an insulated fixture (with a material of polyetheretherketone (PEEK)). The counterweight can realize the horizontal adjustment of the upper tribopair and the weights can be used to achieve each load (1 N to 10 N). The PTFE film as the lower tribopair was attached on the insulated acrylic sheet which was fixed on the linear motor. The reciprocating sliding speed was controlled between 200 and 1000 mm/s. The foam tape between the PTFE film and the acrylic sheet can make a well contact of the ball and plate. The reciprocating sliding displacement was 20 mm, which was two-thirds of the length of the PTFE film. The friction force was detected by the strain gauges and recorded on a computer by the strain amplifier (DPM-951A, Kyowa, Japan) for calculating the friction coefficient. The duration of the friction tests varied between 2400 s and 10,500 s according to the experimental requirements.

Liquid lubricants such as hexadecane, squalane, poly alpha olefins-6 (PAO6), 1-butyl-3-methylimidazolium hexafluorophosphate (BMIMPF6), and silicone oil were applied for the liquid lubrication tests. The main properties of the five liquid lubricants are listed in Appendix A. Graphene nanosheets were purchased from the Xianfeng Nano Company (Nanjing, China). Graphene-doped silicone oil was prepared by mixing graphene nanosheets and silicone oil. Span-80 was used as a surfactant to facilitate a better dispersion of graphene nanosheets into silicone oil. Graphene materials were uniformly dispersed into silicone oil by using an ultrasonic magnetic stirrer for 60 min. The 50 μL of graphene-doped silicone oil with graphene concentration (mass fraction) varied from 0.005 wt% to 0.5 wt% and was carefully dropped to the surface of the PTFE film by using a plastic dropper before the sliding test. All the experiments were conducted under ambient environment with a temperature of 25 ± 3 °C and a relative humidity of 55 ± 10% RH.

The output short-circuit current and open-circuit voltage were measured by using a low-noise current amplifier (SR570, Stanford Research System, Sunnyvale, CA, USA) and an electrometer (6517B, Keithley, Cleveland, OH, USA), respectively. The electric generation mechanism of the reciprocating sliding-mode TENG based on the metallic ball and PTFE film is schematically illustrated in Appendix A. The video of measuring the open-circuit voltage of the PTFE film sliding against a steel ball under the graphene-doped silicone oil lubrication condition is presented in Appendix A. The surface potential of the PTFE film before and after the sliding test was detected by using a digital low-voltage static meter (KSD-3000, Kasuga, Japan).

An optical microscope (LV-150N, Nikon, Tokyo, Japan) was applied to observe the wear scars on the metallic balls after sliding against the PTFE films. A 3D laser confocal microscope (VK-X200, Keyence, Osaka, Japan) was employed to analyze the wear tracks on the PTFE films after friction tests and to estimate the wear rate of the PTFE films. The element compositions of the wear scars on the metallic balls were analyzed by using a scanning electron microscopy (SEM, Scios, FEI, Hillsboro, OR, USA) equipped with an energy dispersive X-ray spectroscope (EDS). The bonding configuration data of the worn surfaces on the wear scars of the steel balls were characterized by using a Raman spectroscope (LabRAM HR Evolution, Horiba, Kyoto, Japan) operating with a laser power of 0.1 mW and a laser wavelength of 532 nm.

## 3. Results

### 3.1. Effect of Liquid Lubrication on the Triboelectrical Performances of PTFE Films

It is well known that liquid lubricants could effectively reduce the friction and wear of sliding contacts [39,40]. Yoo et al. studied the effect of silicone oil on the friction coefficient between a stainless-steel ball and plate, whereby the average friction coefficient greatly decreased from 0.76 to 0.14 with the addition of silicone oil to the contact interface [40]. Therefore, five types of liquid lubricants (hexadecane, squalane, PAO6, BMIMPF6, and silicone oil) were applied to investigate the triboelectrical behaviors of the PTFE films. The optical photographs and corresponding chemical structures of these five liquid lubricants are illustrated in Figure 2a.

Friction curves of the PTFE film sliding against a copper ball under different lubrication conditions with a load of 5 N and a sliding speed of 200 mm/s are shown in Figure 2b. The average friction coefficients and corresponding standard deviations are calculated by taking the friction coefficients of 1800–2400 s and are summarized in Figure 2e. Typical friction curves in one cycle at steady state (i.e., 2300 s) under dry friction and silicone oil lubrication conditions are provided in Appendix A, respectively. Specifically, the friction coefficient in the dry friction condition increased rapidly from 0.13 to 0.22 in the initial 100 s, and then reached a stable value of approximately 0.26 together with a large fluctuation of the friction coefficient. The friction coefficients decreased greatly to less than 0.10 with the addition of five different liquid lubricants to the contact interfaces. Particularly, after adding silicone oil to the sliding contact, the friction coefficient decreased gradually from the start of the test and finally reduced to a steady state of 0.035, less than one-seventh of that in the dry friction condition. This great difference could also be observed in the one cycle friction curves, as shown in Appendix A.

The short-circuit current and open-circuit voltage of the PTFE film running against a copper ball in dry friction and liquid lubrication conditions are shown in Figure 2c and d, respectively. The maximum short-circuit current and maximum open-circuit voltage of each test are shown in Figure 2e. The maximum short-circuit current was determined as the highest value of the measured short-circuit current, and the maximum open-circuit voltage was determined from the largest gap between the highest value and the lowest value of the measured open-circuit voltage in one cycle, which was inconsistent with the previous work [41]. In the case of the dry friction condition, the maximum short-circuit current and maximum open-circuit voltage were 1.24 nA and 0.32 V, respectively. The output triboelectrical performances of the PTFE films were apparently improved under hexadecane, squalane, PAO6, and silicone oil lubrication, except the open-circuit voltage slightly decreased to 0.23 V under BMIMPF lubrication. The highest output short-circuit current and open-circuit voltage of 5.47 nA and 2.40 V were obtained under squalane lubrication, respectively. The output short-circuit current and open-circuit voltage at steady state (i.e., 2300 s) under dry friction and silicone oil lubrication conditions are provided in Appendix A, respectively. Obviously, the short-circuit current and open-circuit voltage under silicone oil lubrication increased by 2.2 times and 3.5 times, respectively, compared to those under the dry fiction condition. These results are consistent with the previous studies, where it has been argued that the electrical output performances of the liquid lubricated-TENGs are highly related to the dynamic viscosity and relative permittivity of the lubricants, whereby low viscosity and low relative permittivity are favorable for achieving high electrical behavior [30,31].

Therefore, it was found that a stable and low-friction coefficient of 0.035, together with an output short-circuit current and an open-circuit voltage of 2.71 nA and 1.12 V, respectively, were achieved under the silicone oil lubrication condition, which suggested great potential for obtaining outstanding triboelectrical performances of the PTFE films.

### 3.2. Effect of Mating Ball Material on the Triboelectrical Performances of PTFE Films

The material selection of tribopair in TENGs is not only crucial for modulating the friction and wear behaviors but is also important for enhancing the output electrical performances (e.g., short-circuit current and open-circuit voltage) of the sliding contacts under lubrication conditions [42,43]. Du et al. [44] proposed a customized bearing type triboelectric nanogenerator (B-TENG) and evaluated the influence of ball material (i.e., steel, copper, and aluminum) on the output triboelectrical performances of the PTFE film-based B-TENGs. Hence, three types of ball (copper, aluminum, and steel) were employed to obtain better triboelectrical performances of the PTFE films-based TENG under the silicone oil lubrication condition. The optical photographs and enlarged optical images of the three types of balls are shown in Figure 3a(I–III) and (IV–VI), respectively. It could be clearly observed that the surface of the steel ball was the smoothest together with the least number of pits, followed by the copper ball, and the surface of the aluminum ball was the roughest with more pits.

Friction curves of the PTFE films sliding against three different balls under the silicone oil lubrication condition with a load of 5 N and a sliding speed of 200 mm/s are shown in Figure 3b. Specifically, the friction coefficient of the PTFE film running against the aluminum ball continuously decreased from 0.11 to 0.06 during the entire sliding test. On the contrary, the friction coefficient of the PTFE film sliding against the copper and steel balls reached a steady state after the initial running-in process. Friction coefficients in steady stage were less than 0.05, which were much lower than that with the aluminum ball. Typical friction curves at steady state (1800 to 2400 s) of the PTFE films sliding against different balls are showed in Figure 3c. The average friction coefficients calculated from the friction data from 1800 to 2400 s are shown in Figure 3f. It could be observed that the lowest average friction coefficient of 0.027 was achieved by using the steel ball. Similarly, Lin et al. [45] studied the friction coefficients between the polished chemical vapor deposition (CVD) films and different metallic balls, and the results indicated that the stainless-steel ball sliding against the diamond film demonstrated a lower friction coefficient than the copper and aluminum ball.

The output short-circuit current and open-circuit voltage of the PTFE film in contact with various metallic balls under the silicone oil lubrication condition are shown in Figure 3d and e, respectively. The corresponding maximum short-circuit current and open-circuit voltage are summarized in Figure 3f. Clearly, the highest short-circuit current and open-circuit voltage of 6.09 nA and 2.23 V, respectively, were obtained for the PTFE film sliding against the aluminum ball. The lowest short-circuit current and open-circuit voltage of 1.64 nA and 0.62 V, respectively, were observed for the PTFE film sliding against the steel ball.

Considering that the steel ball exhibited a lower friction coefficient than that of the copper and aluminum balls, it was selected as the upper tribopair in the following experiments for achieving better friction behaviors of the PTFE films.

### 3.3. Effect of Graphene-Doped Silicone Oil on the Triboelectrical Performances of PTFE Films

Two-dimensional materials such as graphene, MXene, and MoS_2_ have been extensively applied for tuning the triboelectrical performances of the polymer films-based TENGs [42,46,47]. To further improve the tribological behavior of the PTFE film under the silicone oil lubrication condition, graphene-doped silicone oil was prepared by mixing graphene nanosheets with silicone oil at different concentrations, as shown in Figure 4a.

Friction curves of the PTFE film sliding against a steel ball under different graphene-doped silicone oil lubrication conditions with a load of 5 N and a sliding speed of 200 mm/s are shown in Figure 4b. The corresponding average friction coefficients calculated as the average values of the friction coefficients ranging from 1800 to 2400 s of the friction tests are shown in Figure 4e. Generally, the friction coefficient of the PTFE film under the silicone oil lubrication condition decreased with the doping of graphene nanosheets. Especially with the addition of graphene nanosheets even at a mass fraction of merely 0.005 wt%, the friction coefficient decreased sharply after a very short running-in process and reached a steady state with a value of less than 0.020. The lowest average friction coefficient was calculated to be 0.016. With the increase in graphene concentration to 0.01 wt% and 0.1 wt%, the friction coefficient increased (i.e., 0.025 and 0.024), but was still less than that under pure silicone oil lubrication (0.027 at 0 wt%). With the further increase in concentration to 0.5 wt%, a slightly higher friction coefficient (i.e., 0.029) was obtained. Therefore, the addition of graphene nanosheets into silicone oil with a concentration between 0.005 wt% and 0.1 wt% is beneficial for achieving a lower friction coefficient of the PTFE film.

The output short-circuit current and open-circuit voltage of the PTFE film running against a steel ball under the graphene-doped silicone oil lubrication condition are shown in Figure 4c and d, respectively. The corresponding maximum short-circuit current and open-circuit voltage are summarized in Figure 4e. The short-circuit current and open-circuit voltage were 1.64 nA and 0.62 V, respectively, under a pure silicone oil lubrication condition. Both the short-circuit current and open-circuit voltage increased gradually with the increase in graphene concentration from 0.005 wt% to 0.1 wt%. The highest short-circuit current and open-circuit voltage of 2.42 nA and 0.92 V were obtained at the graphene concentration of 0.1 wt%. With the further increase in graphene concentration to 0.5 wt%, the short-circuit current and open-circuit voltage decreased to 0.32 nA and 0.18 V, respectively, which were lower than those obtained under a pure silicone oil lubrication.

It was clarified that the lowest friction coefficient of 0.016 was achieved with the doping of graphene nanosheets into silicone oil at a concentration of 0.005 wt%. Hence, graphene-doped silicone oil with a graphene concentration of 0.005 wt% was determined as the best candidate liquid lubricant for the subsequent experiments.

### 3.4. Effect of Load and Sliding Speed on the Triboelectrical Performances of PTFE Films

To further reduce the low friction coefficient of the PTFE films, the effects of load and sliding speed on the triboelectrical performances of the PTFE film sliding against a steel ball lubricated by graphene-doped silicone oil (graphene concentration of 0.005 wt%) were studied. According to the previous works, the running-in period is an important parameter for achieving a stable super-low friction coefficient of the sliding contacts in engineering fields [48,49,50,51]. Hence, the duration of the friction tests was extended from 2400 s to 4500 s for better exploring the super-low friction behavior of the PTFE film sliding against a steel ball under the graphene-doped silicone oil lubrication condition.

Friction curves of the PTFE film sliding against a steel ball under graphene-doped silicone oil (0.005 wt%) lubrication condition with different loads (1 N to 10 N) and sliding speeds (200 mm/s to 1000 mm/s) are shown in Figure 5a and b, respectively. The corresponding average friction coefficients of the PTFE films are shown in Figure 5e,f, which were calculated as the average values of the friction coefficients from 3600 s to 4500 s of the sliding tests. The friction coefficient fluctuated in a range of 0.017 to 0.025 under the load of 1 N, which was much greater than that under other loads. This phenomenon could also be found at the dry friction condition, as shown in Appendix A, where a larger fluctuation of the friction coefficient was observed at the load of 1 N compared with that under the load of 5 N. With the increase in the load from 1 N to 10 N, the average friction coefficient decreased gradually from 0.022 to 0.012. Moreover, with the increase in the sliding speed from 200 mm/s to 1000 mm/s, the average friction coefficient also decreased gradually from 0.012 to 0.008. Furthermore, it could be found that when lower friction coefficients were obtained, a longer running-in period for achieving a steady state was required. Most importantly, after a 3000 s running-in process, the super-low friction coefficient of less than 0.01 was successfully achieved with the PTFE film sliding against a steel ball under graphene-doped silicone oil (graphene concentration of 0.005 wt%) lubrication at a load of 10 N and a sliding speed of 1000 mm/s.

The output short-circuit current and open-circuit voltage are shown in Figure 5c and d, respectively. The corresponding maximum short-circuit current and open-circuit voltage under various loads and sliding speeds are shown in Figure 5g and h, respectively. Both the maximum short-circuit current and open-circuit voltage increased strongly with the increase in load from 1 N to 10 N. Specifically, the maximum short-circuit current increased from 1.52 nA to 3.13 nA and the maximum open-circuit voltage increased from 0.36 V to 0.99 V, as shown in Figure 5g. Nevertheless, with the increase in the sliding speed from 200 mm/s to 1000 mm/s, the maximum short-circuit current also increased significantly from 3.13 nA to 8.49 nA. Meanwhile, the maximum open-circuit voltage approached a stable stage and fluctuated between 0.91 V and 0.99 V.

The initial surface potential of the PTFE films was controlled to be 0 V at the beginning of each sliding test by using an ion fan. Surface potentials of the PTFE films after sliding tests under different loads and sliding speeds are shown in Figure 5e and f, respectively. Obviously, the surface potential increased approximately two times from −74 V to −144 V with the increase in the load from 1 N to 10 N. However, the surface potential decreased gradually from −144 V to −105 V with the increase in the sliding speed from 200 mm/s to 1000 mm/s. It was revealed that the load was more important than the sliding speed for enhancing the surface potential as well as the electric output performances of the PTFE film in sliding-mode TENGs. Zhang et al. [37] reported an increase in the surface potential of the PTFE film by applying a cyclic normal load (1N-5N-1N), which resulted in a considerable increase in output short-circuit current from 55 nA to 225 nA for the PTFE film-based sliding-mode TENG. Therefore, intensive investigation on the electron excitation effect derived from the load is favorable for attaining higher electrical output of the sliding-mode TENGs.

In order to verify the feasibility of achieving the super-low friction electrification of the PTFE film sliding against a steel ball, the optimum experiment conducted under the graphene-doped silicone oil (graphene concentration of 0.005 wt%) lubrication condition with the load of 10 N and the sliding speed of 1000 mm/s was carefully repeated at an extended sliding duration of 10,500 s, as shown in Figure 6. It was clearly observed that the friction coefficient dropped progressively from 0.03 to approximately 0.01 after a 4500 s sliding process. Most interestingly, a super-low friction coefficient of less than 0.01 was substantially achieved at 6000 s and the average friction coefficient of 0.008 was obtained at the final steady stage. Correspondingly, the maximum short-circuit current and open-circuit voltage at stable state (i.e., 10,000 s) were 9.24 nA and 0.95 V, respectively, as shown in the inset images in Figure 6 which were in good agreement with those in Figure 5. Therefore, it was confirmed that the super-low friction electrification of the PTFE film was obtained under the graphene-doped silicone oil lubrication (0.005 wt%) with the load of 10 N and the sliding speed of 1000 mm/s.

## 4. Discussion

In order to gain more insights into the super-low friction electrification mechanisms of the PTFE film sliding against a steel ball lubricated by the graphene-doped silicone oil, worn surfaces on the steel balls and PTFE films were systematically investigated, as shown in Figure 7. Three representative wear scars on the steel balls with different statuses of friction coefficients were selected. The first one was the steel ball lubricated by the pure silicone oil under 5 N and 200 mm/s for 2400 s with a relative high friction coefficient of 0.027. The second one was the steel ball lubricated by the graphene-doped silicone oil (0.005 wt%) under 5 N and 200 mm/s for 2400 s with a relative low friction coefficient of 0.016. The third one was the steel ball lubricated by the graphene-doped silicone oil (0.005 wt%) under 10 N and 1000 mm/s for 4500 s with a super-low friction coefficient of 0.008.

The wear scars on the steel balls were analyzed by using optical microscopy SEM in combination with EDS mapping, as shown in Figure 7a. It can be seen that the wear scar of the steel ball under pure silicone oil lubrication (μ = 0.027) is greater than that under graphene-doped silicone oil lubrication (μ = 0.016). Interestingly, wear scars could hardly be observed on the steel ball under graphene-doped silicone oil lubrication (μ = 0.008), although higher load and sliding speed and longer sliding period were applied. It seemed that the addition of an appropriate amount of graphene not only reduced the friction coefficient of the PTFE film, but also reduced the wear on the steel ball. In the case of the steel ball lubricated by pure silicone oil (μ = 0.027), a carbon element could only be accumulated on the wear debris from the PTFE film. When graphene was added into silicone oil (μ = 0.016), the intensity of the carbon element increased significantly on the worn surface. Moreover, with the increase in the load and sliding speed (μ = 0.008), more carbon element appeared at the contact area on the sliding interface. The distribution of carbon element on the surface of the steel ball indicated that graphene had successfully entered the sliding interface of the steel ball and PTFE film. Moreover, the oxygen element was uniformly distributed on the worn surface of the steel ball. The silicone element was derived from the silicone oil and was randomly distributed on the worn surface.

The surface morphologies of the PTFE films were analyzed by using the 3D laser confocal microscopy. The 3D images of the wear tracks on the PTFE films are shown in Figure 7b(I–III), and the corresponding cross-sectional profiles are shown in Figure 7b(IV–VI). Specifically, the length and depth of the wear track on the PTFE film with μ = 0.027 were 782.43 μm and 15.24 μm, respectively, which were larger than the wear track on the PTFE film with μ = 0.016. It was in good agreement with the optical microscopy images in Figure 7a. However, the length and depth of the wear track on the PTFE film with μ = 0.008 were increased to 1081.56 μm and 28.37 μm, respectively. The wear volume and wear rate of the PTFE films were derived from the cross-sectional profiles, and the results are shown in Figure 7c. With the addition of graphene nanosheets into silicone oil, the wear volume of the PTFE films decreased from 0.100 mm^3^ to 0.084 mm^3^, and the corresponding wear rate decreased from 15.70 × 10^−5^ mm^3^/Nm to 13.18 × 10^−5^ mm^3^/Nm as well. It could be found that the addition of graphene into the silicone oil enhanced the wear resistance of the PTFE film. With the increase in load and sliding speed from 5 N to 10 N and 200 mm/s to 1000 mm/s, respectively, together with the extension of the sliding period from 2400 s to 4500 s, the wear volume of the PTFE film (μ = 0.008) increased to 0.415 mm^3^, however, the corresponding wear rate was drastically decreased to 8.19 × 10^−5^ mm^3^/Nm, which was approximately half of that lubricated by pure silicone oil. Thus, it was claimed that the lowest wear rate of the PTFE film was achieved together with the super-low friction coefficient.

Figure 7d indicates the Raman spectra (ranging between 500 cm^−1^ and 3500 cm^−1^) of the wear scars on the steel balls. The Raman spectrum of the original graphene nanosheets is also provided for reference. It could be seen that there were no detectable peaks on the Raman spectrum of the steel ball lubricated by pure silicone oil (μ = 0.027). However, the Raman spectra of the other two steel balls lubricated by the graphene-doped silicone oil (0.005 wt%) contained an individual D peak located at 1350 cm^−1^ and G peak located at 1590 cm^−1^, which could strongly prove the appearance of graphene materials on the wear scars. In addition, the shape of the D peak and G peak of the steel ball under 10 N and 1000 mm/s (μ = 0.008) were much higher than the steel ball under 5 N and 200 mm/s (μ = 0.016), which further confirmed the existence of graphene-rich materials on the wear scar. The strong peak located around 2900 cm^−1^ could be attributed to the newly formed C-H absorptions during the sliding test [52].

The super-low friction electrification mechanisms for the PTFE film sliding against a steel ball under the graphene-doped silicone oil lubrication condition are proposed and schematic illustrations are shown in Figure 8. In liquid-solid TENGs, it has been argued that the alternating current was generated from the continuous bouncing motion of the water droplet between the two tribolayer materials [53,54]. Therefore, it was assumed that when there was liquid lubricant between the metallic ball and the PTFE film, the triboelectrification mechanism of the TENG remains identical with that of the solid-solid TENG, and triboelectricity is still generated from the charge transfer of the micro-asperity on the solid surface. On the one hand, the introduction of silicone oil to the sliding contacts would prevent the direct contact between the PTFE film and metallic ball, thus avoiding the severe wear of the contact surface and reduce the friction coefficient from 0.26 in dry friction condition to 0.027 in pure silicone oil lubrication, as shown in Figure 8a. Moreover, with the addition of graphene nanosheets into the silicone oil, transfer film containing graphene was substantially generated on the contact interface, leading to the low shear strength of the sliding contact [55,56,57] and enhancing the silicone oil’s load-carrying capacity as well [23]. Thus, a lower friction coefficient of 0.016 was obtained with the addition of graphene nanosheets at a mass faction of 0.005 wt%, as shown in Figure 4. With the increase in the load and sliding speed to 10 N and 1000 mm/s, respectively, due to the fluid dynamic pressure effect, as shown in Figure 8b, the enhanced flowability of the graphene nanosheets brought more graphene materials into the contact area, which would weaken the bonding strength of the contact interface and further reduce the friction coefficient [58], and hence a super-low friction coefficient of 0.008 was achieved after the running-in process, as shown in Figure 5. The robust superlubric status (as shown in Figure 6) indicated that graphene nanosheets were firmly adhered on the contact interfaces during the extended sliding process.

On the other hand, it has been argued that the existence of the nonpolar lubrication oil (e.g., silicone oil, mineral oil, and soybean oil) could suppress air-breakdown on the sliding interface and improve the surface charges of the tribolayer material, and thus result in higher electrical performances [17,19,30]. Similarly, the output short-circuit current and open-circuit voltage greatly increased for the lubricated-TENGs (e.g., hexadecane, squalane, PAO6, and silicone oil lubrication) compared to the unlubricated TENG, as shown in Figure 2e. Generally, it has been claimed that the formation of transfer film on the contact interface is detrimental for attaining higher electrical output of the TENGs [30]. The introduction of different lubrication oil could prevent the formation of transfer film and thus result in higher electric outputs, as shown in Figure 2. However, with the formation of graphene-containing transfer film on the contact interface by adding graphene nanosheets into the silicone oil, the output short-circuit current and open-circuit voltage basically increased with the increase in the concentration of graphene nanosheets, and the highest short-circuit current and open-circuit voltage of 2.42 nA and 0.92 V were obtained at the graphene concentration of 0.1 wt%. The enhanced electrical performances could be attributed to the channel effect of the graphene nanosheets, where a transfer aisle was provided by the graphene nanosheets to promote the electron migration [38], as schematically shown in Figure 8c. Moreover, with the further increase in the mass fraction of graphene nanosheets to 0.5 wt%, both the short-circuit current and open-circuit voltage decreased to 0.32 nA and 0.18 V, respectively, which were lower than those obtained under pure silicone oil lubrication, as shown in Figure 4. The deterioration of the output electrical performance can be attributed to the particle agglomeration phenomenon of the graphene materials under higher mass fraction, which has been observed in the previous studies [47,59].

Finally, it could be concluded that the synergistic lubricating effect of the silicone oil and graphene nanosheets (solid-liquid coupling [39,60]) not only facilitates the achievement of macro-scale superlubricity of the sliding contact but also favors the achievement of high output electrical performance. Some recent studies on the low friction electrification mechanism of the TENGs are summarized in Appendix A. These comparison results strongly prove the significance of graphene-doped silicone oil lubrication for boosting high-performance TENGs, in terms of excellent friction and wear behavior as well as electrical output, which suggest great potential for versatile TENGs applications.

## 5. Conclusions

In this work, to achieve super-low friction electrification of the TENGs, the triboelectrical behaviors of the sliding-mode TENGs based on the PTFE films and metallic balls under both dry friction and liquid lubrication conditions were investigated by using a customized testing platform with a ball-on-flat configuration. The average friction coefficient of the PTFE films running against a steel ball decreased to 0.027 under a pure silicone oil lubrication condition. It was further decreased to 0.016 when lubricated by silicone oil with the doping of graphene nanosheets at a mass fraction of 0.005 wt%. Most interestingly, a super-low friction coefficient of 0.008 was achieved under graphene-doped silicone oil lubrication with the optimized high load of 10 N and high sliding speed of 1000 mm/s. The corresponding wear rate of the PTFE film was drastically decreased to 8.19 × 10^−5^ mm^3^/Nm, which was approximately half of that lubricated by pure silicone oil. Thus, it was claimed that the ultra-low wear rate of the PTFE film was achieved together with the super-low friction behavior. Simultaneously, the output short-circuit current and open-circuit voltage were enhanced by 6.8 times and 3.0 times, respectively, compared to the dry friction condition. The excellent triboelectrical performances of the PTFE films sliding against steel balls are attributed to the synergistic lubricating effects of the silicone oil and graphene nanosheets. The current research provides valuable insights into obtaining superlubricity of the lubricated-TENGs and paving the way for the design of high-performance superlubric TENGs.

## Figures and Tables

**Figure 1 micromachines-14-01776-f001:**
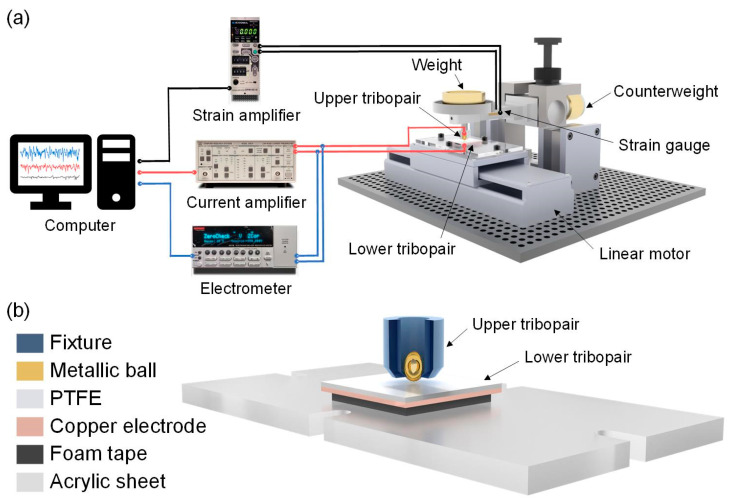
Experimental setup for characterization of the triboelectrical performances of PTFE films. (**a**) Schematic diagram of the self-developed testing platform which could simultaneously measure the friction coefficient, short-circuit current, and open-circuit voltage of the PTFE films running against the metallic balls under the dry friction and liquid lubrication conditions. (**b**) Schematic diagram of the lower and upper tribopair.

**Figure 2 micromachines-14-01776-f002:**
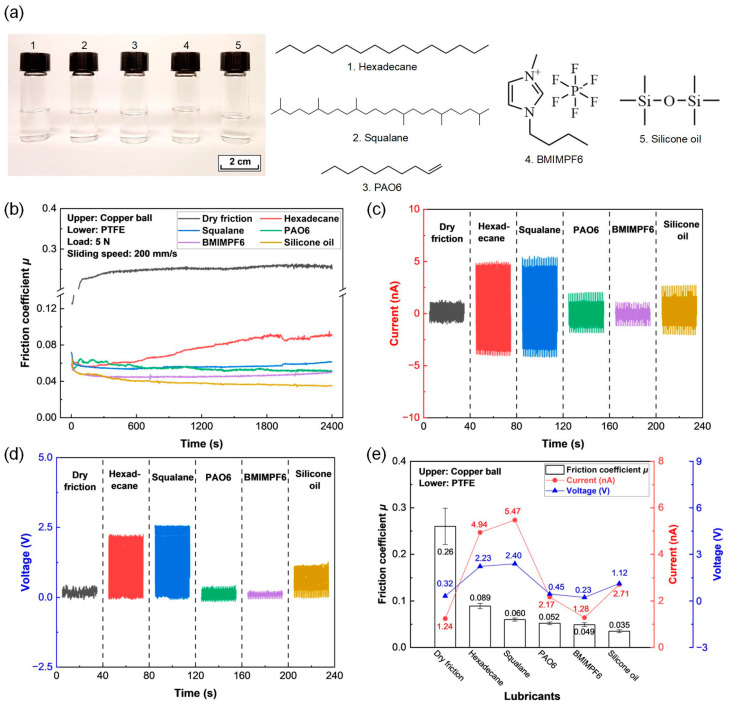
Triboelectrical performances of the PTFE film sliding against a copper ball under dry friction and liquid lubrication conditions with a load of 5 N and a sliding speed of 200 mm/s. (**a**) Optical photographs and corresponding chemical structures of the five liquid lubricants. (**b**) Friction curves. (**c**) Output short-circuit current. (**d**) Output open-circuit voltage. (**e**) Average friction coefficient, maximum short-circuit current, and maximum open-circuit voltage.

**Figure 3 micromachines-14-01776-f003:**
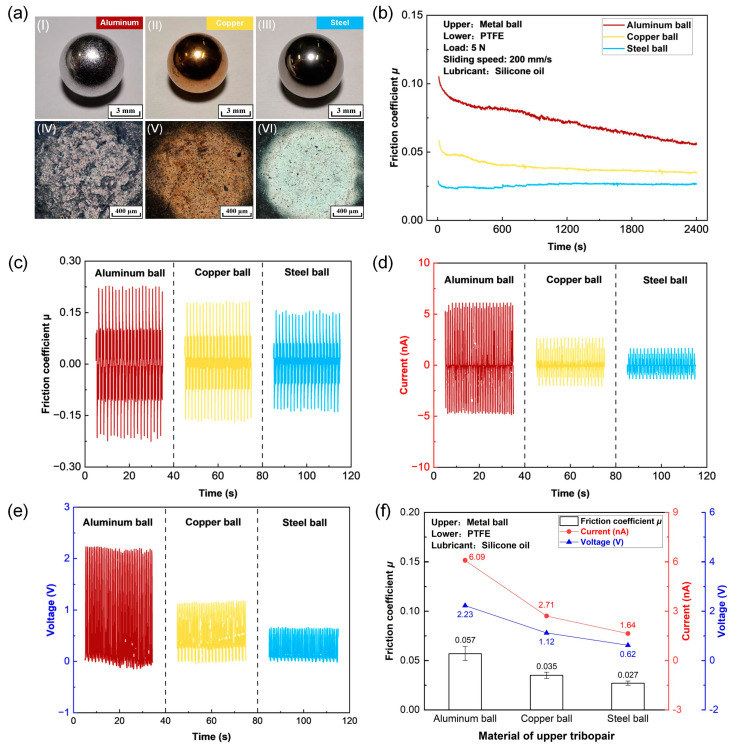
Triboelectrical performances of the PTFE films sliding against three types of metallic balls (aluminum, copper, and steel) under the silicone oil lubrication condition with a load of 5 N and a sliding speed of 200 mm/s. (**a**) Optical photographs (I–III) and corresponding enlarged optical images (IV–VI). (**b**) Friction curves. (**c**) Typical friction coefficients at steady state. (**d**) Output short-circuit current. (**e**) Output open-circuit voltage. (**f**) Average friction coefficient, maximum short-circuit current, and maximum open-circuit voltage.

**Figure 4 micromachines-14-01776-f004:**
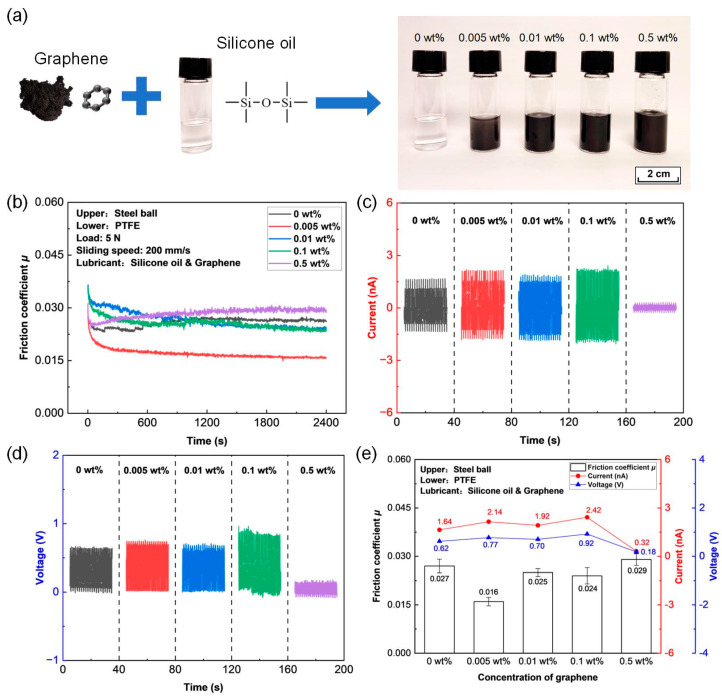
Triboelectrical performances of the PTFE film sliding against a steel ball under graphene-doped silicone oil (graphene concentration of 0–0.5 wt%) lubrication condition with a load of 5 N and a sliding speed of 200 mm/s. (**a**) Optical photographs of silicone oil doped with different concentrations of graphene (0–0.5 wt%). (**b**) Friction curves. (**c**) Output short-circuit current. (**d**) Output open-circuit voltage. (**e**) Average friction coefficient, maximum short-circuit current, and maximum open-circuit voltage.

**Figure 5 micromachines-14-01776-f005:**
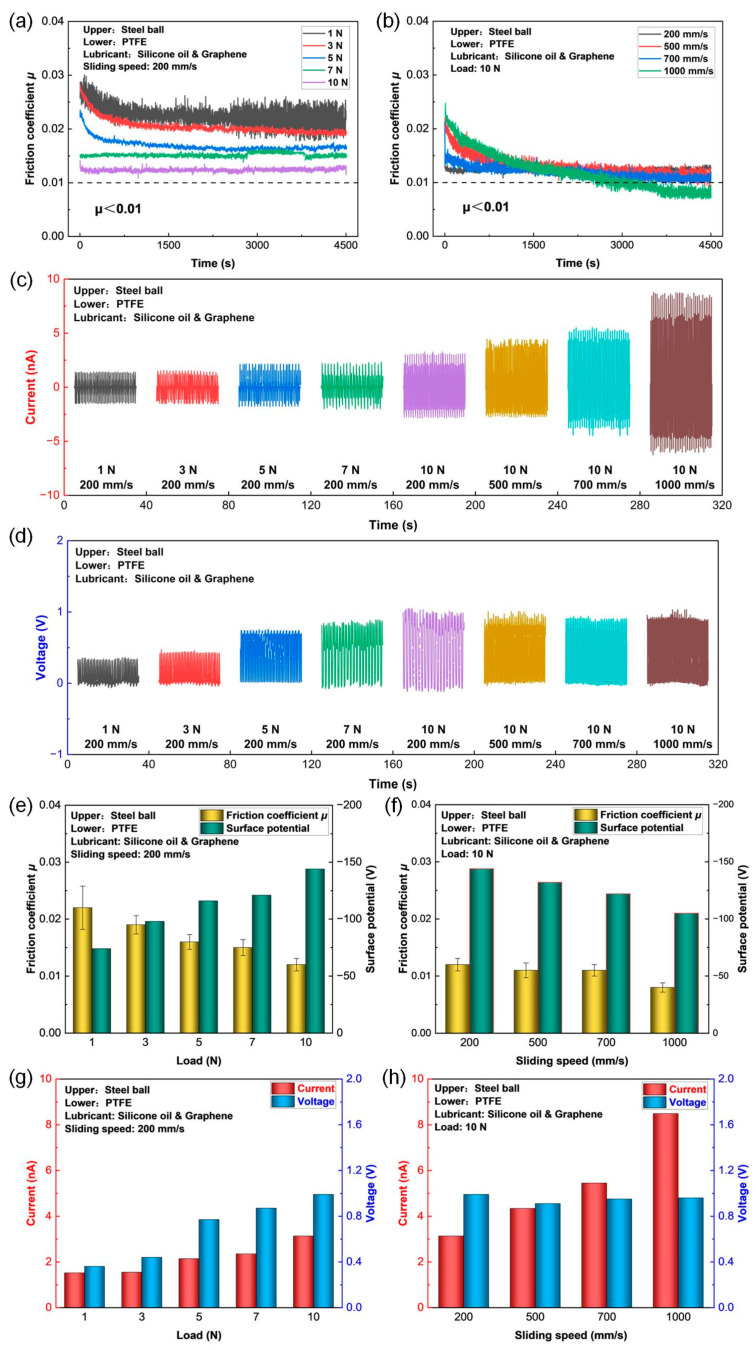
Triboelectrical performances of the PTFE film sliding against a steel ball under the graphene-doped silicone oil (graphene concentration of 0.005 wt%) lubrication condition with different loads (1 N to 10 N) and sliding speeds (200 mm/s to 1000 mm/s). (**a**) Friction curves under different loads with a sliding speed of 200 mm/s. (**b**) Friction curves under different sliding speeds with a load of 10 N. (**c**) Output short-circuit current. (**d**) Output open-circuit voltage. (**e**) Average friction coefficient and surface potential of the PTFE films under different load with a sliding speed of 200 mm/s. (**f**) Average friction coefficient and surface potential of the PTFE films under different sliding speeds with a load of 10 N. (**g**) Maximum short-circuit current and open-circuit voltage of the PTFE films under different loads with a sliding speed of 200 mm/s. (**h**) Maximum short-circuit current and open-circuit voltage of the PTFE films under different sliding speeds with a load of 10 N.

**Figure 6 micromachines-14-01776-f006:**
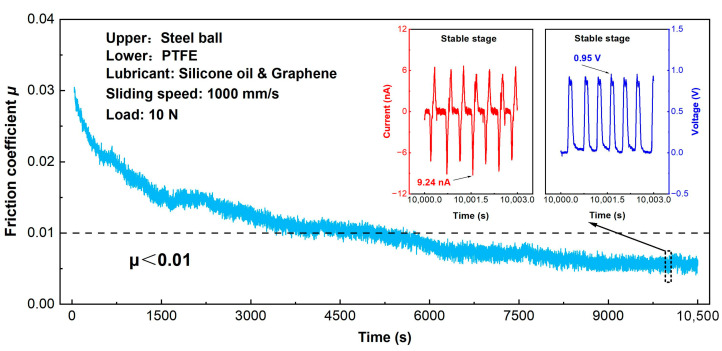
Friction curve of the PTFE film sliding against a steel ball under the graphene-doped silicone oil (graphene concentration of 0.005 wt%) lubrication condition with a load of 10 N, a sliding speed 1000 mm/s, and a duration time of 10,500 s. Insets are the output short-circuit current and open-circuit voltage at steady state (i.e., 10,000 s).

**Figure 7 micromachines-14-01776-f007:**
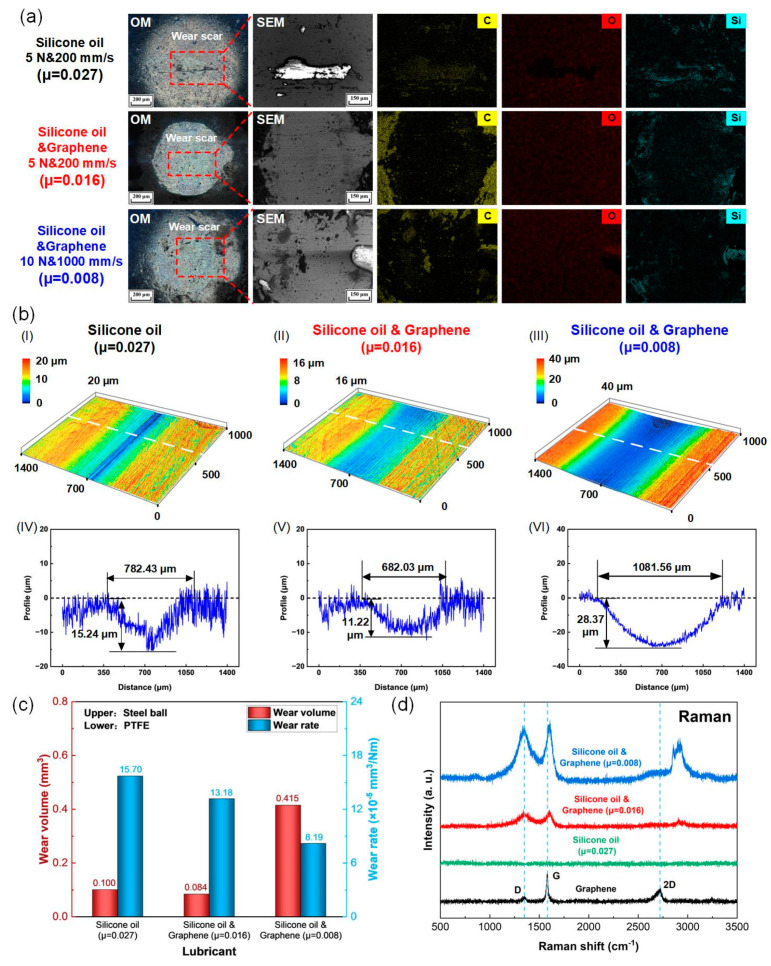
Surface characterizations of the worn surfaces on the steel balls and PTFE films after sliding test under graphene-doped silicone oil lubrication condition with different loads and sliding speeds. (**a**) Optical microscopy images, SEM images, and corresponding elements distributions of the wear scars on the steel balls. (**b**) (I–III) 3D optical images and (IV–VI) corresponding cross-sectional profiles of the wear tracks on the PTFE films. (**c**) Wear volume and calculated wear rate of the PTFE films. (**d**) Raman spectra of the wear scars on the steel balls.

**Figure 8 micromachines-14-01776-f008:**
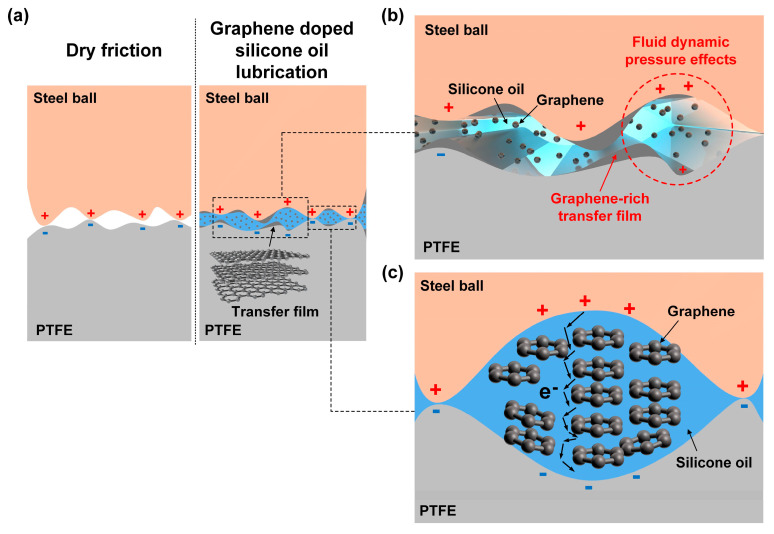
The super-low friction electrification mechanisms for the PTFE film sliding against a steel ball under the graphene-doped silicone oil lubrication condition. (**a**) Macro-scale model for the PTFE film running against a steel ball under dry friction and graphene-doped silicone oil lubrication. (**b**) Super-low friction mechanism. (**c**) Electron generation mechanism.

## Data Availability

The data that support the findings of this study are available from the corresponding author upon reasonable request.

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
