# Peer review of "Super-Low Friction Electrification Achieved on Polytetrafluoroethylene Films-Based Triboelectric Nanogenerators Lubricated by Graphene-Doped Silicone Oil"

_micromachines, 2023, doi:10.3390/mi14091776_

Round 1

Reviewer 1 Report

In this paper, to achieve super-low friction electrification of the TENGs, the tribological and electrical behaviors of the sliding-mode TENGs based on the polytetrafluoroethylene (PTFE) films and metallic balls under both dry friction and liquid lubrication conditions were systematically explored by using a customized testing platform with a ball-onflat configuration.

The title of this paper is the study on the mechanism of excellent triboelectric properties of the prepared materials, which is equivalent to taking a series of analysis equipment to study the materials and analyze the obtained data. The topic and research are relatively normal. However, the experimental process and data are complete, according to the basic mechanism that PTFE films have excellent tribological and electrical properties. This can explain.

As for the biggest highlight of this paper, the super-low friction electrification of TENG is realized by graphene doped silicone oil lubrication. An article "Dissipation Mechanisms and Superlubricity in Solid Lubrication by Wet-Transferred Solution-Processed Graphene Flakes: Implications for Micro Electromechanical Devices" published in ACS APPLIED NANO MATERIALS Journal in 2023 and an article "Physico-chemical and tribological properties of commercial oil-bio-lubricant mixtures dispersed with graphene nanoplatelets" published in RSC ADVANCES Journal in 2023 both appear in lubricating oil doped with graphene nanosheets to improve the tribological properties and mechanical durability of materials. However, compared with the work of these two articles, this work increases the super-low friction electrification of PTFE film and steel ball as the research point. It can be rated as a supplementary innovation point.

In the reference section on figure 3a in line 219 on page 6, a detailed comparison of the magnified optical images of the three materials can be added.

Generally speaking, this article can continue to make further changes to make it more readable, so as to keep its fluent expression.

Reviewer 2 Report

This manuscript discusses the utilization of liquid lubricants for sliding mode TENGs to achieve simultaneous enhancement of both the life span and electrical output. However, the manuscript is not well organized and some of the scientific issues are not thoroughly discussed. Therefore, it is not suitable for publication in its present form.

(1) Similar lubrication techniques for TENG applications have been proposed previously . The new contributions of this paper compared to the previous literature need to be better summarized. For example, comparasion to [35] and [36].

(2) The title is not appropriate, which seems to cover only the content of section 3.3
(3) Solid-solid interface based TENG and solid-liquid interface based TENG involve very different physical mechanisms. Is the fundamental triboelectric mechanism different when there is liquid lubricant between solid-solid interfaces?
(4) For TENG based on solid-liquid interfaces, authors can refer to the following literature:
Advanced Materials Technologies, 2023, 8(2): 2201369.
Nano Energy, 2019, 58: 579-584.
(5) The authors discuss the influence of many factors on TENG. So, is there an optimal design solution? If not, what should be the trade-off?

Reviewer 3 Report

Over all, this manuscript is well written. The authors investigated the super-low friction electrification of the PTFE films based TENGs lubricated by graphene doped silicone oil. They found that the ultra-low wear rate of the PTFE film was achieved together with the super-low friction behavior. Meanwhile, the output short-circuit current and open circuit voltage were also enhanced. They also provide some valuable results. In my opinion, the present version is fine and can be accepted. I also have some comments regarding this study:

1.) The authors built up the experimental installation to test the friction coefficient. I was wondering if it is well calibrated to guarantee the accuracy, or compared the results with other instruments such as Universal Material Tribometer?

2.) The authors select the hexadecane, squalane, PAO6, and silicone oil lubrication to investigate the effect of liquid lubrication on the triboelectrical performances. As far as I know, there are several water base lubricants that can achieve super low friction, such as Phosphoric acid series, ionic liquid and so on. Did they try these liquids?

3.) Can the authors explain how they chose the maximum value of open-circuit voltage? The signal values of voltage for some cases are negative, did the authors consider this condition.

4.) Some details about the experimental setup. From Figure S3, the wire is connected to the ground, how did the authors set up the ground connection? What is the material of the upper tribopair holder, is it insulative?

Reviewer 4 Report

This study is so interesting for publishing after considering some minor corrections:

-          Line 29, please check “21st”

-          Line 32, TENG is already defined in the abstract, no need to define it again here

-          Line 96, can you correct the surface roughness (Ra) subscript

-          I see that the research question of this study should be mentioned in the abstract

-          Figure 3 f, can you enhance its resolution?

-          Figure 7d, Is Raman related to graphene or graphene oxide? And how many layers? This reference may help Reduced graphene oxide-functionalized zinc oxide nanorods as promising nanocomposites for white light emitting diodes and reliable UV photodetection devices - ScienceDirect , https://doi.org/10.1016/j.matchemphys.2023.128063.

-          The working mechanism should be explained better in figure 8

-          This study missing a clear comparison of your results with literature reviews, it is better to put it in the table

-          There are some typing errors in the manuscript, please recheck it

-           

 Moderate editing of English language required

Round 2

Reviewer 2 Report

No more comments.

Reviewer 4 Report

thank you for your report

 Moderate editing of English language required